# Assessment of Treatment Response to Lenvatinib in Thyroid Cancer Monitored by F-18 FDG PET/CT Using PERCIST 1.0, Modified PERCIST and EORTC Criteria—Which One Is Most Suitable?

**DOI:** 10.3390/cancers14081868

**Published:** 2022-04-07

**Authors:** Gundula Rendl, Gregor Schweighofer-Zwink, Stefan Sorko, Hans-Jürgen Gallowitsch, Wolfgang Hitzl, Diana Reisinger, Christian Pirich

**Affiliations:** 1Department of Nuclear Medicine and Endocrinology, University Hospital Salzburg, Paracelsus Medical University Salzburg, Müllner Hauptstr. 48, 5020 Salzburg, Austria; g.schweighofer-zwink@salk.at (G.S.-Z.); diana.reisinger@gmail.com (D.R.); c.pirich@salk.at (C.P.); 2Department of Nuclear Medicine and Endocrinology, PET/CT Centre, Klinikum Klagenfurt am Wörthersee, 9020 Klagenfurt, Austria; stefan.sorko@kabeg.at (S.S.); hans-juergen.gallowitsch@kabeg.at (H.-J.G.); 3Research and Innovation Management, Biostatistics and Publication of Clinical Trial Studies, Paracelsus Medical University Salzburg, 5020 Salzburg, Austria; wolfgang.hitzl@pmu.ac.at; 4Department of Ophthalmology and Optometry, University Hospital Salzburg, Paracelsus Medical University Salzburg, 5020 Salzburg, Austria; 5Research Program Experimental Ophthalmology and Glaucoma Research, University Hospital Salzburg, Paracelsus Medical University Salzburg, 5020 Salzburg, Austria

**Keywords:** F-18 FDG PET/CT, radioiodine-refractory differentiated thyroid cancer, tyrosine kinase inhibitor, lenvatinib, metabolic response

## Abstract

**Simple Summary:**

In patients with progressive metastatic radioiodine-refractory differentiated thyroid cancer, tyrosine kinase inhibitor (TKI) therapy with lenvatinib improves progression-free survival. Early identification of patients with therapeutic responses to the TKI therapy is clinically relevant due to common side effects and toxicity. The use of metabolic response criteria derived from F-18 fluorodeoxyglucose (FDG) positron emission tomography/computed tomography (PET/CT) imaging can help to identify those patients who benefit from treatment. In our study, we evaluated different response criteria, namely, the Positron Emission Tomography Response Criteria In Solid Tumors (PERCIST) and European Organization for Research and Treatment of Cancer (EORTC) criteria, to find out which system might be more reliable and suitable in everyday clinical practice. The EORTC criteria could be applied to all patients, and the different PERCIST criteria, in 80% and 88%, respectively. Regarding their survival and treatment responses, patients with a progressive disease could be reliably identified by using all the criteria.

**Abstract:**

Background: We aimed to compare the established metabolic response criteria PERCIST and EORTC for their applicability and predictive value in terms of clinical response assessment early after the initiation of lenvatinib therapy in patients with metastatic radioiodine-refractory (RAI) thyroid cancer (TC). Methods: In 25 patients treated with lenvatinib, baseline and 4-month follow-up F-18 FDG PET/CT images were analyzed using PERCIST 1.0, modified PERCIST (using SUVpeak or SUVmax) and EORTC criteria. Two groups were defined: disease control (DC) and progressive disease (PD), which were correlated with PFS and OS. Results: PERCIST, mPERCIST, PERCISTmax and EORTC could be applied in 80%, 80%, 88% and 100% of the patients based on the requirements of lesion assessment criteria, respectively. With PERCIST, mPERCIST, PERCISTmax and EORTC, the patients classified as DC and PD ranged from 65 to 68% and from 32 to 35%, respectively. Patients with DC exhibited a longer median PFS than patients with PD for EORTC (*p* < 0.014) and for PERCIST and mPERCIST (*p* = 0.037), respectively. Conclusion: EORTC and the different PERCIST criteria performed equally regarding the identification of patients with PD requiring treatment changes. However, the applicability of PERCIST 1.0 using SULpeak seems restricted due to the significant proportion of small tumor lesions.

## 1. Introduction

The outcome of differentiated thyroid carcinoma (DTC), contributing to around 90% of all thyroid cancer cases [1], is favorable in most patients, including initial treatment with surgery, followed by radioactive iodine (RAI) remnant ablation and suppression of thyroid-stimulating hormone (TSH) using thyroxine therapy. However, local recurrence or distant metastases can emerge in approximately 15% of these patients during follow-up, requiring subsequent RAI therapies. In two-thirds of these DTC patients, RAI-refractory disease will develop [2]. In patients with RAI-refractory disease, the 10-year survival rate is around 10% and the mean life expectancy declines to about 3–5 years [3,4].

Multi-targeted tyrosine kinase inhibitor (TKI) therapy can be considered an important treatment option for DTC patients with RAI-refractory disease [5]. Sorafenib, lenvatinib and most recently cabozantinib have been approved by the Food and Drug Administration (FDA) and the European Medicines Agency for the treatment of thyroid cancer in RAI-refractory disease [6,7]. Their use relies on significant prolongations of median progression-free survival (PFS) compared to a placebo of 18.3 vs. 3.6 months for lenvatinib [8], 10.8 vs. 5.8 months for sorafenib [9] and 11 vs. 1.9 months for cabozantinib [10].

The impact of F-18 fluorodeoxyglucose (FDG) positron emission tomography/computed tomography (PET/CT) has been widely demonstrated for staging and response assessment in thyroid cancer [11,12]. The early identification of patients with therapeutic response to TKI therapy is clinically relevant, in order to identify those patients who are unlikely to benefit from the current therapy regimen, which can be associated with substantial, though manageable, toxicity [13,14,15,16].

The concept and impact of interim FDG PET/CT after therapy induction has been evaluated in different solid tumors [17,18,19]. Wahl et al. [20] provided structured guidance for response assessment using FDG PET/CT in solid tumors, described as the Positron Emission Tomography Response Criteria In Solid Tumors (PERCIST 1.0). Another system to assess treatment response in solid tumors has been developed by the European Organization for Research and Treatment of Cancer (EORTC) [21].

Two recent studies assessed metabolic tumor response in patients undergoing lenvatinib therapy using FDG PET/CT [22,23]. Valerio et al. showed in a collective of 33 patients that FDG PET/CT could predict the response to therapy after 4 weeks of lenvatinib, whereby overall survival (OS) was correlated with metabolic response [22]. Ahmaddy et al. showed in 22 patients that tumor response assessment using modified (m)PERCIST outperformed morphological response assessment by CT using the Response Evaluation Criteria in Solid Tumors 1.1 (RECIST 1.1) [23]. In this study, metabolic responders according to mPERCIST exhibited a significantly higher median PFS and disease-specific survival (DSS).

Therefore, the question arises of whether the metabolic criteria PERCIST 1.0, modified PERCIST and EORTC, which rely on different criteria and segmentation parameters, i.e., SUVmax, SUVpeak and SULpeak, are always applicable and valuable in RAI-refractory TC. The aim of this study was to compare these validated response criteria for predicting clinical response early in the course of lenvatinib treatment in patients with progressive RAI-refractory TC. Therefore, we assessed the applicability and compared the prognostic value of three metabolic response criteria: PERCIST 1.0, modified PERCIST and EORTC.

## 2. Materials and Methods

### 2.1. Study Population

In total, 25 patients (12 women and 13 men) with advanced metastatic RAI-refractory thyroid cancer were included in this retrospective two-center study. Of these, 15 and 10 patients were included from the departments of Nuclear Medicine and Endocrinology at University Hospital Salzburg (Salzburg, Austria) and Klinikum Klagenfurt am Wörthersee (Klagenfurt, Austria), respectively. The following inclusion criteria were applied: 1. Metastatic, RAI-refractory TC; 2. Documented clinical progression of disease over the past 6 months; 3. Treatment with lenvatinib based on consensual multidisciplinary endocrine tumor board recommendation; 4. Baseline and follow-up FDG PET/CT imaging study available. FDG PET/CT imaging has been the standard of care for metastatic thyroid cancer patients at both nuclear medicine institutions for more than 10 years.

The leading ethics committee of the federal state of Salzburg approved the retrospective data analysis (Ethikkommission Salzburg Nr. 1376/2018 from 26 July 2018), and the study was conducted in accordance with the Declaration of Helsinki, as well as national and international guidelines. The need to obtain informed consent was waived due to the retrospective design of the study.

All patients on lenvatinib treatment underwent routine 1-to-3-month follow-up studies for the evaluation of symptoms, blood pressure and vital signs, laboratory testing for thyroglobulin (TG), thyroid-stimulating hormone (TSH), thyroglobulin antibodies (TgAb), and urinary analysis, respectively.

Routine follow-up FDG PET/CT imaging for detection of disease progression was fixed at 3–4 month intervals, and after 1 year of therapy at 4-to-6-month intervals. For our retrospective analysis, we reviewed the FDG PET/CT data at the initiation of lenvatinib treatment and the first follow-up FDG PET/CT imaging after the start of lenvatinib therapy using the same PET scanner and the same protocol in each patient.

### 2.2. F-18 FDG PET/CT Imaging

In the Department of Nuclear Medicine and Endocrinology in Salzburg, all 15 patients underwent fasting FDG PET/CT imaging with a low-dose or diagnostic CT as clinically indicated, after fasting for at least 6 h prior to the exam.

PET images were acquired 60 min after injection of an intended activity of 4 MBq F-18 FDG per kg of bodyweight using an Ingenuity TF (Philips Healthcare, PC Best, The Netherlands) PET/CT scanner from the skull to the mid-thigh in 3D mode with 50% overlap and 75 s per bed position. Prior to PET acquisition, a low dose CT acquisition for attenuation correction and anatomical correlation was performed with the same longitudinal field of view. In one patient (pat. no 1), a contrast-enhanced CT (100 mL Visipaque with a flow of 2 mL/s and 90 s delay) was clinically indicated. PET images were reconstructed on a 144 × 144 matrix, using an iterative algorithm (BLOB-OS-TF). CT images were reconstructed on a 512 × 512 matrix using the usual Filtered Back Projection algorithm. After reconstruction, the final slice thicknesses were 4 and 3 mm for PET and CT, respectively.

In the PET/CT Centre of the Department of Nuclear Medicine and Endocrinology in Klagenfurt, all patients underwent F-18 FDG PET/CT with a contrast-enhanced CT protocol. Patients were required to fast for at least 6 h prior to PET/CT imaging. PET images were acquired 60 min after FDG injection (range: 200 to 370 MBq) using a Biograph 64 mCT flow motion (Siemens Healthcare, Erlangen, Germany) PET/CT scanner from the skull to the mid-thigh in 3D mode with various scan velocities (dynamic speed: head: 1.4 mm/s; cervical region: 1 mm/s; chest 1.4 mm/s; abdomen and pelvis: 0.8 mm/s). Prior to PET acquisition, a diagnostic CT acquisition for attenuation correction and the morphological diagnosis was performed with the same longitudinal field of view.

All scans were performed using a contrast-enhanced CT (60 mL Accupaque for patient weights less than 60 kg, 80 mL for patient weights between 60 and 100 kg and 100 mL for patient weights over 100 kg, with a flow of 3 mL/s and 30 s delay).

PET images were reconstructed on a 200 × 200 matrix using an iterative algorithm (TrueX + ToF, 4 iterations and 21 subsets, Gaussian filter 3.0). CT images were reconstructed on a 512 × 512 matrix using the sinogram-affirmed iterative reconstruction (SAFIRE) algorithm. After reconstruction, the final slice thicknesses were 2 and 3 mm for PET and CT, respectively.

### 2.3. PET Imaging Interpretation

Index lesions were selected at baseline, defined as the lesion with the most intense uptake in different compartments (local recurrence in thyroid bed, lymph node, visceral, muscular, or osseous metastases) and were evaluated in the follow-up FDG PET/CT by experienced, board-certified nuclear medicine physicians using IntelliSpace Portal software (IntelliSpace Portal Tumor Tracking, Phillips 2015). The following metabolic parameters were derived through segmentation of all index tumor lesions of the different compartments: maximum/mean and peak standardized uptake value (SUVmax, SUVmean, SUVpeak) and peak standardized uptake value adjusted to body surface area (SULpeak). The metabolic tumor volume (MTV) was delineated using a computer-assistant tool to outline the margins of the tumor volume automatically and following manual modification by using axial, coronal and sagittal image projections, respectively. The total lesion glycolysis (TLG) was calculated as MTV × SUVmean.

### 2.4. FDG PET/CT Response Evaluation by PERCIST 1.0 and EORCT

Changes in the uptake value of the index lesion in all compartments were measured per patient at baseline and in follow-up PET/CT. The single hottest lesion represents the target tumor lesion with the highest uptake.

Four categories of metabolic treatment response were used as defined for both PERCIST 1.0 and EORTC: complete metabolic response (CMR), partial metabolic response (PMR), stable metabolic disease (SMD) and progressive metabolic disease (PMD) (Table 1).

We applied either SUVpeak for modified PERCIST (mPERCIST), as described by Michl et al. [24], or SUVmax (PERCISTmax) by the method presented here, adapting the PERCIST 1.0 criteria [20] originally relying on SULpeak as a PET segmentation parameter.

In agreement with clinical decision making based on the response to therapy, CMR, PMR and SMD were categorized as disease control (DC) and patients with PMD as progressive disease (PD).

### 2.5. Statistical Methods

The data were checked for consistency. Kaplan–Meier analyses were applied to compare disease control and progressive disease groups. Means were compared using bootstrap *t*-tests, variances were tested with Levene’s test and data were tested for normality. The log-rank randomization test and bootstrap *t*-tests were performed using 5000 Monte Carlo simulations. Results were illustrated using box–whisker plots. All reported tests were two-sided, and *p*-values < 0.05 were considered statistically significant. All statistical analyses in this report were performed by use of NCSS (NCSS 10, NCSS, LLC. Kaysville, UT, USA), STATISTICA 13 (Hill, T. and Lewicki, P. Statistics: Methods and Applications. StatSoft, Tulsa, OK, USA).

## 3. Results

### 3.1. Patient Characteristics

We included 25 patients (12 women and 13 men) with advanced metastatic RAI-refractory thyroid cancer undergoing lenvatinib treatment at the Department of Nuclear Medicine and Endocrinology at University Hospital Salzburg (Salzburg, Austria; n = 15) and Klinikum Klagenfurt am Wörthersee (Klagenfurt, Austria; n = 10), respectively.

Most patients had follicular thyroid carcinoma (FTC, 64%) and were initially diagnosed with stage pT3 (56%) and pT2 (16%), respectively.

All patients had advanced metastatic RAI-refractory disease, mostly with pulmonary (92%) and lymph-node metastases (80%) after multiple local as well as systemic therapies prior to lenvatinib (Table 2). The mean age of patients at initial diagnosis was 60 ± 13 years (median age 64 years, range 33–78 years) and 67 ± 12 years (median age 71 years, range 39–81 years) at the beginning of lenvatinib treatment.

Nineteen patients (76%) started with the recommended daily dose of 24 mg of lenvatinib daily. Two (8%) and four more patients (16%) started with reduced dosages of 10 mg or 14 mg lenvatinib daily, respectively, due to a poor clinical condition (ECOG 2) or comorbidities (arterial hypertension).

The mean PFS of all the subjects was 31.9 ± 20.3 months (median 16.1 months; range 6.4–65.1 months) and the mean overall survival (OS) was 34.6 ± 19.6 months (median 32.7 months; range 6.8–65.1). During follow-up, 12 patients (48%) died, of whom 7 (28%) were treated with lenvatinib until death.

Data analysis comprised FDG PET/CT imaging data of 25 scans before initiation of lenvatinib treatment and 25 scans at first follow-up. The mean time from baseline FDG PET/CT to treatment initiation was 0.8 ± 1.1 months, and the first follow-up FDG PET/CT was performed after a mean of 4.3 ± 1.5 months.

### 3.2. Response Rates with mPERCIST, PERCIST 1.0, PERCISTmax and EORTC Criteria

With mPERCIST, PERCIST 1.0, PERCISTmax and EORTC, 13 out of 20, 13 out 20, 15 out of 22, and 17 out of 25 patients were classified as DC, respectively. Table 3 details the proportion of patients with CMR, PMR and SD, respectively. Furthermore, eight patients applying EORTC criteria and seven patients applying all PERCIST criteria had PD.

Table 4 provides an overview of all the response criteria with corresponding percentage changes in respective SUV.

### 3.3. Response Rate with mPERCIST, PERCIST 1.0 and PERCISTmax Criteria Based on Selection of Hottest or All Lesions

For mPERCIST and PERCIST 1.0, five patients (20%) were excluded from both analyses. In three patients, PET data was not comparable according to the PERCIST 1.0 criteria, and in another two patients, no valid SULpeak assessment could be made.

Using mPERCIST with a solitary assessment of the single hottest lesion demonstrated DC in 13 out of 20 patients (65%), and PD in seven patients (35%), whereas the analysis including all tumor lesions revealed that 14 patients (70%) showed DC and six patients (30%) PD, respectively.

With the hottest vs. all lesion approaches, PMR was seen in five (25%) vs. seven patients (35%) and PMD in seven (28%) vs. six patients (24%), respectively. Two patients were classified as CMR (10%) and five as SMD (25%) in both analyses.

The comparison of mPERCIST data with a selection of either hottest or all lesions revealed one case of major (treatment-relevant) change in classification from PD to DC (pat. no. 1), whereas minor changes (from PMR to SMD or SMD to PMR), which are considered as not relevant for patient management, could be found in three patients (12%).

Applying PERCIST 1.0 with the selection of either all lesions or the hottest lesion, 13 out of 20 patients (65%) had DC, and seven patients (35%) had PD. Again, five patients (20%) could not be classified; three because they had non-comparable PET study data, as defined by PERCIST 1.0, and in two patients (10%), SULpeak was not measurable correctly.

CMR could be seen in two patients (10%), PMR in six patients (30%), SMD in five patients (25%) and PMD in seven patients (35%) with both PERCIST approaches. Minor changes, not relevant for treatment decisions, could be found in two patients (10%) when comparing the analyses employing the hottest lesion vs. all lesions.

With PERCISTmax based on hottest or all lesion analyses, 15 out of 22 patients (68%) showed DC, and 7 out of 22 patients (32%) PD. Furthermore, two, eight, five and seven patients were classified as CMR (8%), PMR (32%), SMD (20%) and PMD (28%), respectively, in hottest and all lesion analyses (Table 3). Minor changes could be seen in two patients (8%) when comparing PERCISTmax hottest versus all lesion analyses.

Three patients (12%) were excluded from PERCISTmax analysis because PET data sets were not comparable according to the response criteria. The difference between baseline and follow-up SUVmean in the reference liver exceeded 20%.

### 3.4. Treatment Response with EORTC Criteria Based on Hottest or All Lesions

EORTC revealed DC in 17 out of 25 patients (68%) and PD in 8 out of 25 patients (32%) for both analyses based on the hottest or all lesions. Using hottest-lesion vs. all-lesions approaches, CMR and PMD were seen in two (8%) and eight patients (32%) each, while the frequencies of PMR and SMD were different, with 11 (44%) vs. 10 patients (40%) and 4 (16%) vs. 5 patients (20%), respectively.

Minor changes between EORTC analysis based on the hottest lesion or all lesions could be seen in three patients (12%).

### 3.5. Comparison of Treatment Response Using (Modified) PERCIST(max) 1.0 Hottest Lesion and EORTC Hottest Lesion

The classification of all patients was only possible with the EORTC hottest lesion. Up to five patients were excluded from analysis with the PERCIST 1.0 criteria, mainly attributable to a non-measurable SULpeak at baseline and non-comparable baseline and follow-up PET study data.

CMR was seen in two patients in all classifications. PMR was seen in 11 patients with EORTC, but only 8, 7 and 5 with PERCISTmax, mPERCIST and PERCIST 1.0, respectively.

PMD was seen in eight patients with EORTC, and seven with PERCIST 1.0, PERCISTmax and mPERCIST. This was due to patient no. 10, who was excluded from analysis due to the significant difference in liver FDG uptake, exceeding 20% SUL/SUVmean between the baseline and follow-up PET data set. With EORTC, the patient was could still be classified.

Notably, no major shift from DC to PD classification using the different systems was observed.

### 3.6. Course of PET Segmentation Parameters

The evaluation of single PET parameters, such as SUVpeak, SUVmax and SULpeak, as well as MTV and TLG, showed significant differences in all parameters comparing the decline in PET parameters in DC and PD patients (Table 4, Figure 1a,b).

### 3.7. Progression-Free and Overall Survival According to Metabolic Response (DC versus PD)

The median survival was 32.7 months (range 6.8 to 65.1 months); seven patients (28%) died on lenvatinib therapy (Table 5), and lenvatinib was withdrawn in five patients (20%) who died during the follow-up period. The remaining 13 patients (52%) are still alive and under lenvatinib treatment, except for two patients who underwent a change in treatment due to the progression of disease under lenvatinib.

Using PERCIST 1.0 and mPERCIST based on the hottest lesion, patients with DC exhibited a longer median PFS than patients with PD in the hottest lesion (35.6 vs. 24.9 months, *p* = 0.037) (Figure 2a). With PERCISTmax based on the hottest lesion, the PFS was significantly (*p* = 0.03) longer in patients with DC (median PFS not reached for responders) than in patients with PD (median 24.9 months) (Figure 2b). With EORTC based on the hottest lesion, patients with DC had highly significantly (*p* = 0.014) longer median PFS than patients with PD (58.4 vs. 24.9 months) (Figure 2c).

## 4. Discussion

In this study, we analyzed the applicability and clinical value of four different metabolic response criteria (PERCIST 1.0, mPERCIST, PERCISTmax and EORTC) derived from pretherapeutic and 4-month interim FDG PET/CT imaging in RAI-refractory thyroid cancer patients undergoing therapy with lenvatinib.

PERCISTmax, PERCIST 1.0 and mPERCIST performed equally, with an agreement of 100% for the identification of PD in eligible patients. However, in this small cohort, 20% of the patients were not evaluable using PERCIST 1.0 or mPERCIST. This is in contrast to the EORTC criteria, which were applicable to all 25 patients. There was complete agreement in the identification of PD when comparing EORTC and PERCIST criteria. Previous studies did not report on whether PERCIST 1.0 criteria were applicable to all patients [22,23]. Thus, at least the results from our patient population suggest the use of EORTC criteria based on SUVmax, and applying the same, baseline hottest lesion per region as a practicable approach in clinical routine.

In more detail, limitations to the applicability of PERCIST 1.0 criteria are even more common when based on SULpeak or SUVpeak rather than SUVmax. Thus, PERCISTmax allowed for the assessment of response in 22 out of 25 patients and was therefore not applicable to 12%, while the analysis was restricted to 20 patients with PERCIST 1.0 and mPERCIST and therefore not applicable to 20%.

While there was overall agreement in the proportion of patients with PD by all metabolic criteria, small but clinically non-relevant shifts between the groups of patients classified as having a metabolically stable disease or as displaying partial metabolic response were observed in patients with DC. More notably, both patients achieving complete metabolic responses were identified by all response criteria.

Overall, the response rate of about 65–68% in our patient group fits results from randomized controlled trials with conventional response assessment (SELECT) [8].

Our findings on the metabolic response by FDG PET imaging and their translation into clinical patient management by two distinct patient groups, those with PD or DC, support the concept and clinical use of early metabolic response evaluation by EORTC and PERCIST [20,21] in RAI-refractory TC treated with the TKI lenvatinib [22,23]. The study by Ahmaddy et al. demonstrated that mPERCIST was superior to morphological RECIST 1.1 in a collective of 22 patients [23]. Valerio et al. demonstrated the added value of PERCIST 1.0 in terms of providing early information on response to lenvatinib treatment in 33 patients with RAI-refractory thyroid cancer, where a response rate of 58% was reported [22].

Both studies employed FDG PET/CT imaging and did not report on the number of patients excluded from analysis due to invalid data when applying any of the PERCIST 1.0 criteria. In our study, this proportion of patients was notable (12 to 20%). Though all patients exhibited extensive and advanced disease, there might be various limitations to the use of PERCIST 1.0 criteria. In PERCIST 1.0, the background definition for target lesions is detailed, and SULpeak must be at least 1.5 times greater than the SULmean + 2SDs of a 3 cm spherical region of interest in the normal right liver lobe. In rare cases, the hottest tumor lesion cannot be measured due to the fact that SULpeak in the hottest tumor lesion is lower than the cut-off of the defined background in the liver. For PERCIST 1.0, the target lesion is the most metabolically active tumor with the highest (hottest) FDG uptake in each scan. This means that the target lesion is not the same at baseline and follow-up but the hottest one in each.

In clinical routine, SUVmax is still the most commonly used parameter to assess the metabolic activity of tumor cells. SUVmax can be applied for EORTC criteria. However, SUVmax does not represent the whole tumor burden and is sensitive to changes in weight, etc. [25]. Therefore, SUV corrected by lean body mass (SUL) is recommended in obese patient groups [26]. The SULpeak relating to the average value within a fixed-size region of interest in the hottest part of the tumor is commonly employed for PERCIST 1.0 analysis. Correction by lean body mass avoids artificial high organ SUVs in obese patients, as fatty tissues have a much lower FDG uptake than organ tissue. In our collective, seven patients had a BMI > 30 at baseline, and only four of them did at follow-up, in concordance with the known weight loss under lenvatinib therapy [8]. However, four out of these seven patients were categorized as PMR and one as PMD using all the PERCIST and EORTC criteria. Clinically minor changes from PMR (all PERCIST 1.0 criteria) to SMD (EORTC) could be seen in pat. no. 3, and from SMD (all PERCIST 1.0 criteria) to PMR (EORTC) in pat. no. 12.

In contrast to EORTC, PERCIST 1.0 criteria precisely define target lesions [20,27] with an uptake of at least 1.5 times greater than SULmean plus two standard deviations of a 3 cm spherical region of interest in the normal right lobe of the liver.

For clinical use, EORTC might prove more practicable, since firstly, both SUVmax and SUVmean of the target lesion can be applied, and secondly, no requirements in terms of the number of target lesions are given. Therefore, metastases at any site (lymphatic, visceral and bone) are subject to response evaluation.

Thirdly, EORTC criteria do not define a minimum background, which allows for the inclusion of small tumor lesions with a low or moderate FDG uptake in response assessment. Notably, one patient in our study presented with multiple small pulmonary metastases, eligible for EORTC criteria only.

The results of our study support the use of the hottest lesion approach, which might reflect best the most biologically active lesion within a thyroid tumor, which commonly demonstrates intraindividual heterogeneity [28], and among which the coexistence of different grades of differentiation is well-known [29]. Furthermore, different mechanisms of tumor-cell escape mechanisms have been demonstrated under TKI treatment [27,30].

To the best of our knowledge, this is the first study to compare RECIST 1.0 and EORTC criteria in DTC patients. A similar approach has been published for breast cancer imaging, stating comparable findings for tumor response assessment with PERCIST 1.0 and EORTC [31]. Our findings suggest that the use of the hottest lesion is at least equally effective, and might be more appropriate and feasible for metabolic tumor imaging in clinical routine.

Our study assessed metabolic response after an average of 4 months of lenvatinib therapy. The assessment of metabolic response using either of the metabolic classification criteria provided long-term prognostic information that DC was associated with a median PFS of 58.4 months using EORTC criteria, relating very well to the long-term findings of the extended SELECT study [32]. FGFR is inhibited by lenvatinib and might contribute to preventing the development of long-term resistance [33].

The volume-related PET parameters MTV and TLG have been shown to yield prognostic information in patients with metastatic RAI-refractory DTC [34,35]. We found them to also be significantly different in patients with DC and PD, with *p* = 0.013 and *p* = 0.004, respectively. However, the courses of both parameters were not related to PFS and OS, which is consistent with the analysis by Ahmaddy et al. [23].

Limitations: Firstly, our study was small and retrospective in nature, including a heterogeneous patient population in terms of age and tumor histology and with variations in follow-up frequencies owing to different clinical and logistical needs. However, our patient population was comparable with the Austrian RELEVANT [36] and the international SELECT trials [8] in terms of the extent of disease, the pattern of metastasis and tumor burden.

Secondly, our study employed imaging data from two centers with slight differences in acquisition protocols. However, all patients routinely underwent follow-up examinations with the same protocol on the same PET scanner.

Thirdly, starting dosages of lenvatinib were different in a few cases, and lower doses applied might reduce the therapeutic efficacy of lenvatinib and its metabolic assessment. Notably, dose reduction was also required in the majority of patients in clinical trials such as SELECT due to toxicity and tolerability.

## 5. Conclusions

Overall, EORTC and different PERCIST 1.0 criteria perform equally regarding the identification of PMD patients requiring clinically relevant treatment changes. All metabolic response criteria obtained 4 months after treatment initiation provide valid and distinctive long-term prognostic information. EORTC criteria relying on SUVmax performed at least equally to PERCIST 1.0 criteria and might be a more practical approach in clinical routine due to more appropriate tumor lesion selection. The hottest-lesion approach is sufficient and at least equally predictive for response assessment compared to the analysis including all lesions.

The feasibility and applicability of PERCIST 1.0 using SULpeak seem restricted due to the significant proportion of small lesions commonly found in metastatic TC.

## Figures and Tables

**Figure 1 cancers-14-01868-f001:**
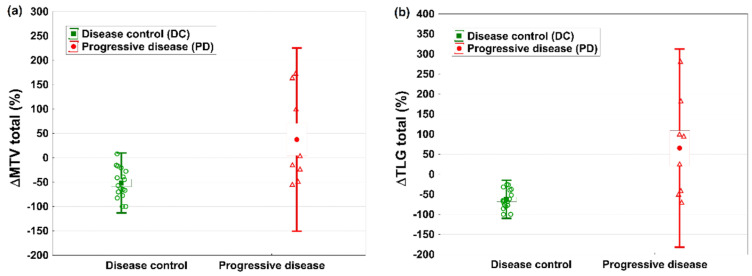
Mean changes in PET parameters (**a**) metabolic tumor volume (MTV) and (**b**) total lesion glycolysis (TLG) in patients with DC and PD according to EORTC; EORTC = European Organization for Research and Treatment of Cancer; DC = disease control; PD = progressive disease.

**Figure 2 cancers-14-01868-f002:**
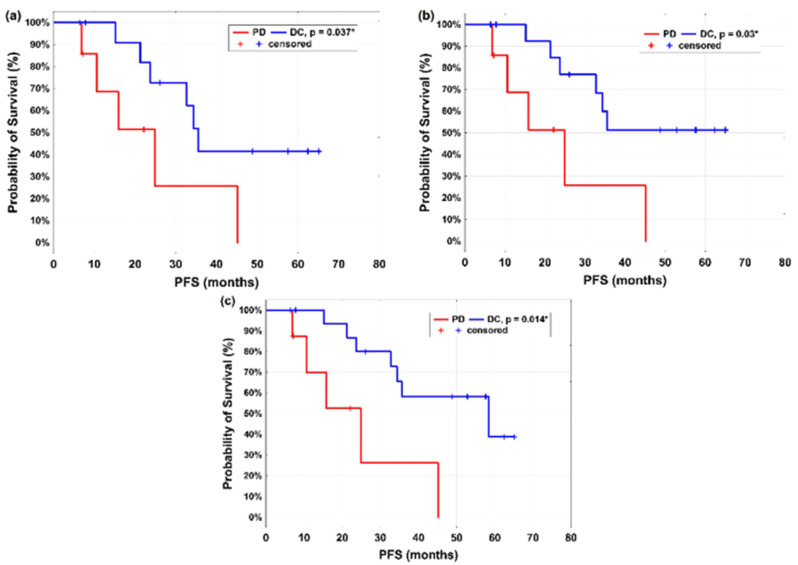
Kaplan–Meier estimate of PFS stratified as DC or PD by metabolic response according to (**a**) PERCIST 1.0 and mPERCIST, (**b**) PERCISTmax. and (**c**) EORTC (hottest lesion); PFS = progression-free survival; DC = disease control; PD = progressive disease; PERCIST 1.0 = PET Response Criteria in solid Tumors; EORTC = European Organization for Research and Treatment of Cancer; mPERCIST = modified PERCIST using SUVpeak; PERCISTmax = modified PERCIST using SUVmax.

**Table 1 cancers-14-01868-t001:** Response classification by EORTC and PERCIST 1.0 criteria.

Response Classification	EORTC ^1^	PERCIST 1.0 ^2^
Progressive metabolic disease (PMD)	Increase in SUVmax of at least 25%	Increase in SULpeak of at least 30% and an absolute increase by at least 0.8 SUL units of the target lesion
	Increase in the extent of FDG uptake (>20% in longest diameter)	Increase in target lesion size by 30%
	New FDG avid lesion(s)	New FDG avid lesion(s)
Stable metabolic disease (SMD)	Increase in SUVmax < 25%	Increase in SULpeak < 30%
	or decrease < 25%	or decrease < 30%
Partial metabolic response (PMR)	Decrease in SUVmax of at least 25%	Decrease in SULpeak of at least 30%
		Decrease of at least 0.8 SUL units
Complete metabolic response (CMR)	Complete resolution of FDG uptake in all lesions (indistinguishable from surrounding normal tissue)	Indistinguishable FDG uptake from surrounding background and SULpeak lower than liver

^1^ EORTC = European Organization for Research and Treatment of Cancer. ^2^ PERCIST 1.0 = PET Response Criteria in solid Tumors.

**Table 2 cancers-14-01868-t002:** Patient characteristics.

Patient Characteristics	N = 25
Median age (years, range)	71 (39–81)
Female—n (%)	12 (48)
Histologic subtype—n (%)	
Follicular (FTC)	16 (64)
Papillary (PTC)	6 (24)
Poorly differentiated	2 (8)
Anaplastic with differentiated component	1 (4)
Metastatic lesions—n (%)	
Lung	23 (92)
Bone	13 (52)
Lymph node	20 (80)
Liver	6 (24)
Brain metastases	2 (8)
Local recurrence	8 (32)
Prior radioiodine therapy—n (%)	
None	0 (0)
One	6 (24)
Two	4 (16)
Three	5 (20)
Four	6 (24)
More than four	4 (16)
Median cumulative radioiodine activity (MBq, range)	16,750 (3610–41,277)
Prior local radiation therapy—n (%)	13 (52)
Prior TKI therapy—n (%)	
Yes	4 (16)
No	21 (84)

**Table 3 cancers-14-01868-t003:** Response classification by mPERCIST, PERCIST 1.0, PERCISTmax and EORTC—Comparison of analysis of the single hottest lesion (hottest lesion) vs. lesions in all compartments (all lesions) EORTC = European Organization for Research and Treatment of Cancer; PERCIST 1.0 = PET Response Criteria in solid Tumors; mPERCIST = modified PERCIST using SUVpeak; PERCISTmax = modified PERCIST using SUVmax; CMR = complete metabolic response (green); PMR = partial metabolic response (yellow); SMD = stable metabolic disease (orange); PMD = progressive metabolic disease (red); n.a. = SUVpeak/SULpeak not available; n.c. = PET studies not comparable.

Nr.	PERCIST 1.0 Hottest Lesion	PERCIST 1.0 All Lesions	mPERCIST Hottest Lesion	mPERCIST All Lesions	PERCISTmax Hottest Lesion	PERCISTmaxAll Lesions	EORTC Hottest Lesion	EORTC All Lesions
1								
2								
3								
4								
5								
6								
7								
8								
9								
10	n.c.	n.c.	n.c.	n.c.	n.c.	n.c.		
11								
12								
13								
14								
15								
16								
17								
18								
19	n.a.	n.a.	n.a.	n.a.				
20	n.a.	n.a.	n.a.	n.a.				
21	n.c.	n.c.	n.c.	n.c.	n.c.	n.c.		
22	n.c.	n.c.	n.c.	n.c.	n.c.	n.c.		
23								
24								
25								

**Table 4 cancers-14-01868-t004:** Mean changes of PET parameters in patients classified as disease control (DC) and progressive disease (PD) according to all PERCIST and EORTC criteria; DC = disease control; PD = progressive disease; PERCIST 1.0 = PET Response Criteria in Solid Tumors; EORTC = European Organization for Research and Treatment of Cancer; mPERCIST = modified PERCIST using SUVpeak; PERCISTmax = modified PERCIST using SUVmax.

Mean Change ± SD (%)	SUVpeak (mPERCIST ^1^)	SULpeak (PERCIST 1.0 ^2^)	SUVmax (PERCISTmax ^3^)	SUVmax (EORTC ^4^)	MTV (cm³) (EORTC ^4^)	TLG (EORTC ^4^)
DC	−46 ± 25	−43 ± 27	−44 ± 31	−42 ± 32	−52 ± 31	−62 ± 24
PD	92 ± 34	103 ± 34	106 ± 19	105 ± 19	38 ± 94	65 ± 124
*p*-value	<0.000001	<0.000001	<0.000001	<0.000001	0.013	0.004

^1^ DC = 13 and PD = 7, ^2^ DC = 13 and PD = 7, ^3^ DC = 15 and PD = 7, ^4^ DC = 17 and PD = 8.

**Table 5 cancers-14-01868-t005:** Number of deaths during lenvatinib treatment depending on metabolic response according to all PERCIST criteria and EORTC (hottest lesion).

	PERCIST 1.0 Hottest Lesion	mPERCIST Hottest Lesion	PERCISTmax Hottest Lesion	EORTC Hottest Lesion
CMR	0	0	0	0
PMR	2	2	3	3
SMD	1	2	1	1
PMD	3	3	3	3

EORTC = European Organization for Research and Treatment of Cancer; PERCIST 1.0 = PET Response Criteria in solid Tumors; mPERCIST = modified PERCIST using SUVpeak; PERCISTmax = modified PERCIST using SUVmax; CMR = complete metabolic response; PMR = partial metabolic response; SMD = stable metabolic disease; PMD = progressive metabolic disease.

## Data Availability

The presented data in this study are available upon request from the corresponding author.

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
