# Peer review of "Assessment of Treatment Response to Lenvatinib in Thyroid Cancer Monitored by F-18 FDG PET/CT Using PERCIST 1.0, Modified PERCIST and EORTC Criteria—Which One Is Most Suitable?"

_cancers, 2022, doi:10.3390/cancers14081868_

Round 1
Reviewer 1 Report
The authors present a prospective study of patients undergoing treatment with lenvatinib for thyroid cancer. The followed response with PET and evaluated responses with the PERCIST and EORTC criteria. They found both useful
Author Response
Dear reviewer (1),
Thank you for your appreciation of our work and manuscript.
Yours sincerely
Gundula Rendl MD PhD
Reviewer 2 Report
The authors compared different methods to evaluate the therapeutic response to lenvatinib by F-18 FDG PET/CT images. It is an important question and due to the current development of more specific target therapies against driver mutations may result in the earlier change of the therapy and clinical benefit to the patient. However, the paper requires improvement from many aspects. My critical comments:
The title is too complicated and did not express the main message of the work.
The simple summary is not acceptable in the present form as it did not contain any relevant information about the work, it is just an introduction.
The abstract also should focus on the main message. It should be emphasized that the PERCIST criteria couldn’t be applied to the 20% of patients. The clarification why it couldn’t be used is also crucial as most of the readers are not nuclear medicine specialists. It is obvious that only 20 patients’ data should be used for the comparison if the analysis was not possible in the remaining five cases.
It cannot be concluded in the abstract that EORTC and PERCISTmax provided the highest predictive value for PFS if it was not demonstrated in the paper.
Anyway, the authors should decide what they would like to compare:
PERCIST 1.0, mPERCIST, PERCISTmax and EORTC and use the terminology consistently.
They mention that the tumor markers, thyroglobulin and anti-TG antibody titers were also measured. The reviewer would be interested in how the tumor markers changed during the treatment and how these lab results correlated with the outcome.
I recommend building the discussion around the main message of the work:
“The use of EORTC criteria relying on SUVmax performed at least equally to PERCIST criteria and might be a more practical approach in clinical routine due to more appropriate tumor lesion selection”
In conclusion, it is a valuable work, but the manuscript needs significant improvement.
Reviewer 3 Report
The authors present a small series of patients from one centre with RAI resistent differetiated thyroid cancer and their apllication of two differet guidelines criteria for treatment and follow up evaluation.
It is an original and practically important study.
The series is quite small. I understand that the phenomenon is rare, but would be a possibility to collect data from several centres an option?
Would heat map contain also abbreviations to be easily followed?
Simple summary would contain also conclusions of the study.
Author Response
Dear reviewer (3),
Thank you for your comments.
- With due respect, we used data from 2 centres, as given in the methods section 2.1.
- Abbreviations were ordered more clearly in the description of Table 3.
- As suggested, we modified the simple summary.
Yours sincerely
Gundula Rendl MD PhD
Round 2
Reviewer 2 Report
The authors corrected all the required items, answered the open questions.